# Performance Testing of *Bacillus cereus* Chromogenic Agar Media for Improved Detection in Milk and Other Food Samples

**DOI:** 10.3390/foods11030288

**Published:** 2022-01-21

**Authors:** Eva Fuchs, Christina Raab, Katharina Brugger, Monika Ehling-Schulz, Martin Wagner, Beatrix Stessl

**Affiliations:** 1Unit of Food Microbiology, Institute of Food Safety, Food Technology and Veterinary Public Health, University of Veterinary Medicine Vienna, 1210 Vienna, Austria; fuchs_e@hotmail.com (E.F.); christina.raab@hotmail.com (C.R.); Martin.Wagner@vetmeduni.ac.at (M.W.); 2Unit of Veterinary Public Health and Epidemiology, Institute of Food Safety, Food Technology and Veterinary Public Health, University of Veterinary Medicine Vienna, 1210 Vienna, Austria; katharina.brugger@vetmeduni.ac.at; 3Functional Microbiology Group, Institute of Microbiology, University of Veterinary Medicine Vienna, 1210 Vienna, Austria; Monika.Ehling-Schulz@vetmeduni.ac.at; 4Austrian Competence Center for Feed and Food Quality, Safety and Innovation (FFOQSI GmbH), 3430 Tulln an der Donau, Austria

**Keywords:** *Bacillus cereus* group, food safety, chromogenic media, performance testing, toxin gene profiling, *panC* sequencing

## Abstract

In this study, the performance of four alternative selective chromogenic *B. cereus* agar was compared to the reference mannitol-yolk polymyxin (MYP) agar (ISO 7932) using inclusion and exclusion test strains (*n* = 110) and by analyzing naturally contaminated milk and other food samples (*n* = 64). Subsequently, the *panC* group affiliation and toxin gene profile of *Bacillus cereus senso lato* (*s.l.*) isolates were determined. Our results corroborate that the overall best performing media CHROMagar™ *B. cereus* (93.6% inclusivity; 82.7% exclusivity) and BACARA^®^ (98.2% inclusivity, 62.7% exclusivity) are more sensitive and specific compared to Brilliance™ *B. cereus*, MYP and ChromoSelect *Bacillus* Agar. Both media allow unequivocal detection of *B. cereus* with low risks of misidentification. Media containing ß-D-glucosidase for the detection of presumptive *B. cereus* may form atypical colony morphologies resulting in a false negative evaluation of the sample. Naturally contaminated samples presented high numbers of background flora, while numbers of presumptive *B. cereus* were below the detection limit (<10 CFU g^−1^ or mL^−1^). Recovery after freezing resulted in the highest detection of *B. cereus s.l*. on BACARA^®^ (57.8%), CHROMagar™ *B. cereus* (56.3%) and MYP agar (54.7%). The *panC*/toxin profile combination IV/A was the most abundant (33.0%), followed by III/F (21.7%) and VI/C (10.4%). More *panC* and toxin combinations were present in 15.6% of samples when reanalyzed after freezing. In order to improve detection and confirmation of *B. cereus s.l.* in food samples, we recommend the parallel use of two complementary selective media followed by molecular characterization (e.g., *panC* typing combined with toxin gene profiling). When determining psychrotolerant or thermophilic members of the *B. cereus* group, the selective agar media should additionally be incubated at appropriate temperatures (5 °C, ≥45 °C). If high-risk toxin genes (e.g., *ces* or *cytK-1*) are detected, the strain-specific ability to produce toxin should be examined to decisively assess risk.

## 1. Introduction

*Bacillus senso latu* (*s.l.*) consists of Gram-positive, rod-shaped, aerobic or facultative anaerobic spore-forming bacteria that are widespread in the environment and commonly found in soil, plant material and in the gut of insects [1,2,3]. As toxin producers and food spoiling bacteria, they pose a health risk and cause economic damage when entering and persisting in the food chain [4,5,6].

The *B. cereus s.l.* group is represented by *B. cereus s.s.*, *B. anthracis*, *B. cytotoxicus*, *B. mycoides*, *B. pseudomycoides*, *B. thuringiensis*, *B. toyonensis* and *B. weihenstephanensis* [2,7,8,9,10].

In order to protect consumer health, a process criterion for presumptive *B. cereus* in infant formulas (≤500 colony forming units [CFU] g^−1^) was set within the Commission Regulation (EC) No 1441/2007 (EC, 2007; https://eur-lex.europa.eu/legal-content/EN/TXT/HTML/?uri=CELEX:32007R1441&from=EN; accessed on 17 December 2021). In addition, warning values are available, for example, for dried herbs and spices; tofu and bakery products (4 log CFU g^−1^); or fruits and vegetables, cereals, pasta, mayonnaises, dressings, soups and ready-to-eat instant products (3 log CFU g^−1^) (https://www.dghm-richt-warnwerte.de/de/dokumente; accessed on 17 December 2021).

In current practice, presumptive *B. cereus s.l.* is detected and enumerated on classical culture media as for example mannitol egg yolk polymyxin (MYP) agar. Chromogenic reactions rely on enzymatic cleavage (e.g., by β-D-glucosidase) of a particular substrate and the release of a chromogen, which is more specific than conventional microbiological growth media. Some chromogenic media additionally detect PLC activity in order to facilitate unambiguous identification. Selectivity is achieved in both media types by the addition of antibiotic substances (e.g., polymyxin B or trimethoprim), which inhibit the growth of undesirable Gram-positive and Gram-negative bacteria [11,12].

Apart from *B. cereus s.l.* counts, strain-specific properties such as toxin gene profiles or other virulence factors need to be investigated for risk characterization efforts. A broad range of phenotypical (e.g., biochemical profile, growth behavior and σ-endotoxin crystal staining) and genotypical methods (e.g., Multilocus Sequence Typing (MLST) and *panC*-typing) are required to accurately identify and group *B. cereus s.l.* on the species level [13,14,15], rendering species differentiation difficult under routine laboratory conditions.

Fourier transform infrared (FTIR) spectroscopy and Matrix-Assisted Laser Desorption/Ionization Time of Flight (MALDI-TOF) Mass Spectrometry have been developed to speed up the identification and characterization of *B. cereus s.l.* and cereulide. However, database richness is decisive for accurate species identification, and an enhanced cloud-based exchange of spectral data would be necessary for propagation [16,17,18,19,20].

As many innovative methods are established exclusively in expert’s laboratories, there is still the need for rapid and unambiguous isolation and differentiation methods applicable in food and dairy plant laboratories. Contemporary chromogenic media may represent a useful tool to facilitate identification of *B. cereus s.l.* and accelerate the time to result by easier visual evaluation of morphology and color changes of media.

This study was initiated to assess and compare the performance of the ISO standard medium MYP agar with four alternative chromogenic selective plating media for detection and enumeration of food-intoxication and spoilage-associated *B. cereus* group members by using a bacterial test strain panel and analyzing naturally contaminated samples under everyday conditions. Furthermore, an in-depth molecular-biological characterization of inclusivity test strains and sample isolates was performed to explore strain-specific features.

## 2. Materials and Methods

### 2.1. Performance Testing of Selective Media

#### 2.1.1. Test Media

Within the scope of this study, the performance of commercially available chromogenic selective media ChromoSelect *Bacillus* (HI; Merck KgaA, Darmstadt, Germany; formerly branded HiCrome™ *Bacillus*), CHROMagar™ *B. cereus* (CH; CHROMagar, Paris, France), Brilliance™ *B. cereus* (BRI; Thermo Fisher Scientific Inc., Oxoid, Waltham, MA, USA) and BACARA^®^ agar (BA; *B. cereus* Rapid Agar; bioMérieux, Marcy l’Etoile, France) was evaluated in comparison to the ISO recommended standard medium MYP [21] (Thermo Fisher Scientific Inc., Oxoid, Waltham, MA, USA). Information on media composition—as indicated in the media manufacturer’s specifications—is listed in Appendix A.

#### 2.1.2. Inclusivity and Exclusivity Test Strains

In order to evaluate the performance of *B. cereus* selective media, a bacterial test strain panel (*n* = 220) consisting of *B. cereus* target organisms (for inclusivity testing, *n* = 110) and non-target *Bacillus* spp. (for exclusivity testing, panel included spoilage-associated microbes and Gram–positive and Gram–negative competitors; *n* = 110) was compiled. *B. cereus s.l.* strains originated from emetic and diarrheal outbreaks (Institute for Microbiology strain collection, University of Veterinary Medicine Vienna), environmental samples, fruits and vegetables, cereals, fish, tea, herbs, spices, milk and dairy products (isolate collection Unit of Food Microbiology, University of Veterinary Medicine Vienna; Appendix A). Exclusivity strains were selected according to their relevance and frequency as food contaminants and covered, among others, isolates deriving from fruits and vegetables, meat and meat products, dried spices and seeds, milk products and dairy processing environments (Appendix A). All strains are preserved as cryogenic cultures (Corning, VWR, Vienna, Austria) in a volume of 1.5 mL brain heart infusion broth (BHI; Merck KGaA, Darmstadt, Germany) with 15% glycerol (Merck KGaA) at −80 °C (GFL Gesellschaft für Labortechnik GmbH, Großwedel, Germany) in the strain collection of the Unit of Food Microbiology.

After activation of test strains from glycerol stocks and subculturing on trypto-casein-soy agar plus 0.6% yeast (TSA-Y; Biokar Diagnostics, Beauvais, France), selective media were inoculated. In order to obtain a few well-defined bacterial colonies, an isolated single colony from the working culture was transferred onto selective *B. cereus* media by fractioned three loop inoculation. After incubation (Ehret GmbH & Co. KG, Emmendingen, Germany) at the specified conditions (Appendix A), the presences of bacterial growth and colony morphology were recorded for all media. By qualitative classification into “typically growing,” “atypically growing”, or “non-growing” strains, media benefits and limitations were determined.

#### 2.1.3. Naturally Contaminated Food Samples

In order to evaluate media reliability under routine laboratory conditions, food samples (*n* = 64) from 18 producers were collected from the production chain and retail level. Food samples (20.3%, *n* = 13/64; producer A–F) belonged to the source categories “fruits and vegetables”, “nuts, nut products and seeds”, “fish and fishery products”, “herbs” and “cocoa and cocoa preparations, coffee, and tea” (Appendix A). Milk samples (79.7%, *n* = 51/64; producer G-R) were heat treated, except for one raw milk sample (bactofugation) provided by a local dairy (Appendix A).

Important information regarding processing was gathered, including the type of milk with reference to animal species (cow or small ruminant), agricultural system (organic or conventional farming), processing and predicted shelf-life (homogenized, pasteurized or high pasteurized) (Appendix A). The majority of milk samples (80.4%, *n* = 41/51) were produced organically. Milk samples were examined after 24 h provocation at 30 °C for enrichment to ensure detection of *B. cereus s.l.* All food samples were analyzed before and after freezing at −20 °C to trigger outgrowth of spores.

In order to prepare sample homogenates, 25 mL or 25 g of food product was diluted 1:10 in sterile buffered peptone water (BPW; Fisher Scientific Inc., Oxoid); food samples were additionally mixed for 180 s in a paddle blender (Stomacher^®^; Seward Ltd., West Sussex, UK). Ten-fold serial dilutions in sterile Ringer’s solution (B. Braun Melsungen AG, Melsungen, Germany) were plated in duplicate up to 10^−5^ on selective agar media by using the spatula method. Following incubation, growth was assessed, and colonies displaying characteristic morphology were enumerated to determine the extent of *B. cereus s.l.* contamination. Randomly picked colonies with typical and atypical morphology were isolated, subjected to confirmation and characterized with regard to *panC* group affiliation and toxin gene profile.

### 2.2. Molecular and Phenotypical Characterization

#### 2.2.1. DNA-Extraction

Bacterial DNA was extracted from *B. cereus s.l.* cultures grown overnight on TSA-Y at 30 °C (Biokar Diagnostics and Merck KGaA) using the Chelex^®^ 100 resin (Bio-Rad Laboratories, Inc., Hercules, CA, USA) method as described by Walsh et al. [22]. After extraction, 100 µL DNA of each isolate was kept at −20 °C until use in characterization experiments.

#### 2.2.2. Confirmation of Group Affiliation

*Bacillus cereus s.l.* strains from culture collections and presumptive isolates from naturally contaminated food products and milk were confirmed as group members by PCR method targeting the gyrase B gene (*gyrB*) as described by Dzieciol et al. [23].

#### 2.2.3. Toxin Gene Screening and Profiling

Confirmed *B. cereus s.l.* strains were screened for their toxin gene content by conventional PCR assays. Amplification was performed according to Ehling-Schulz et al. [24] with minor modifications, addressing the most widespread toxin genes. Two genes of the NHE-complex and two genes of the HBL-complex were taken into consideration: the enterotoxin genes *nheA, nheB*, *hblA* and *hblD*. Furthermore, PCR pre-screening assay were applied for *cytK*-*1/cytK*-*2.* Detection of the emetic toxin cereulide gene *ces* was performed after Dzieciol et al. [23] with slight adjustments. Strain-specific toxin gene profiles were assigned based on prevailing toxin gene combinations as in Ehling-Schulz et al. [24] (Figure 5, Appendix A).

#### 2.2.4. Partial *panC* Sequencing

Amplification, purification and sequencing (LGC, Berlin) of a fragment of the pantothenate synthetase (*panC*) gene were conducted as previously reported [13]. In order to assign *B. cereus s.l.* strains to one of the seven major phylogenetic groups (i.e., I-VII) defined by Guinebretière et al. [13,25], sequences were matched with deposited sequences in a web-based database (https://www.tools.symprevius.org/Bcereus/english.php, accessed on 17 December 2021) (Figure 5, Appendix A).

#### 2.2.5. Assessment of Hemolytic Activity

β-hemolytic activity of inclusivity test strains was determined on Columbia agar plates containing 5% sheep blood (COS; bioMérieux) after overnight incubation at 30 °C [21] (Appendix A).

### 2.3. Evaluation Criteria and Statistics

In order to differentiate the phenotypic appearance of test strains and evaluate their potential for misidentification, the growth of inclusivity and exclusivity strains was classified in typical and atypical according to their reaction(s) and colony morphology on selective media (Figure 1).

A mosaic plot was used for visualizing the results of growth and phospholipase C reactions of inclusivity and exclusivity test strains for each of the tested media MYP, HI, BRI, CH and BA (Figure 1). Detectability of *B. cereus s.l.* in naturally contaminated samples was illustrated in a bar plot (Figure 5). The relative frequency of *panC* group (II–VI) and toxin gene profile (A–F) combinations among *B. cereus s.l.* isolates (*n* = 106) associated with naturally contaminated samples was depicted as pie chart (Figure 6). Graphics were created with open-source statistical computer environment R version 4.1.0 [26].

## 3. Results

### 3.1. Inclusivity and Exclusivity Testing

The detailed strain properties of the inclusivity test strains are presented in Appendix A. The majority of inclusivity test strains (*n* = 110) were assigned to toxin profile C (*nhe*+/*hbl*+; 33.6%, *n* = 37) and A (*nhe*+/*hbl*+/*cytK*+; 27.3%, *n* = 30). The *ces* gene was present in six (5.5%) test strains derived from foodborne outbreaks. Among the target test strains, *panC* group III (30.9%, *n* = 34), IV (30.0%, *n* = 33) and VI (20.0%, *n* = 22) were the most abundant. The most frequent combination of *panC* group and toxigenic profile in the entire panel of inclusivity test strains was IV/A (21.8%, *n* = 24), obtained from milk and dried products (such as tea, spices and mushrooms). Other common combinations were VI/C (17.3%, *n* = 19) isolated from milk, soil and salad, as well as III/D (10.9%, *n* = 12) mainly detectable in strains isolated from protein-rich food (e.g., feta, dried fish and mushrooms).

Examination of target strains showed >99% inclusivity on all media (*n* = 109–110/110); one *B. pseudomycoides* strain did not grow on three selective media (Figure 1). The highest rates of atypical β-D-glucosidase negative colonies were observed on BRI (12.7%, *n* = 14)*,* HI (6.4%, *n* = 7) and CH (5.5%, *n* = 6), resulting in an atypical white phenotype (Figure 2 and Appendix A). Such colony morphologies were largely related to the milk- or soil-derived *panC*-type/toxin profile VI/C. On chromogenic media (CH and BA), the PLC reaction was more distinct in comparison to MYP agar (Figure 2).

Best performing media in terms of exclusivity (Appendix A) were CH (82.7%, *n* = 91/110) and BA (62.7%, *n* = 69/110). Several non-target organisms were not effectively suppressed by polymyxin B in MYP and (82.7%, *n* = 91/110) and HI (88.2%, *n* = 97/110) (Figure 3 and Appendix A). Comparatively low inhibition of exclusivity strains (*n* = 110) was also observed on BRI (60.9%, *n* = 67), although we only observed colony morphologies that could not be misidentified as presumptive *B. cereus* due to their atypical pin-point growth (Figure 3).

PLC reaction typical for the target organisms was observed in three and two exclusivity tests strains on MYP and BA, respectively (*Listeria monocytogenes*, *Paenibacillus polymyxa* and *Serratia marcescens*).

### 3.2. Naturally Contaminated Samples

Milk (*n* = 51) and food (*n* = 13) samples analyzed were contaminated with presumptive *B. cereus* at the limit of detection, resulting in quantitative data below 10 and 100 CFU g^−1^, respectively. Further details on sample characteristics can be found in Appendix A. Typical and atypical *B. cereus s.l.* colonies grown on selective media test panel are shown in Figure 4.

Figure 5 shows the *B. cereus* group containing samples with respect to the distribution of the *panC* group in combination with toxin profiles. The samples were negative in PCR confirmation of the emetic toxin gene (*ces*); in consequence, the toxin profiles B (*nhe, hbl, ces* gene combination positive) and E (*nhe* and *ces* gene combination positive) were not detected. The *panC*/toxin profile combination IV/A was the most abundant in the sample set (33.0%), followed by III/F (21.7%) and VI/C (10.4%). Representatives of *panC* group IV are described as highly cytotoxic and do generally grow at temperatures ≥10 °C. Toxin profile A represents *nhe, hbl* and *cytK* gene (*cytK*-*2*) positive isolates. The enterotoxin genes *nhe, hbl* and *cytK*-2 are located in the chromosome of different species of the *B. cereus* group, whereas the *cytK*-*1* gene is harbored exclusively by thermophilic species *B. cytotoxicus* (*panC* group VII). Representatives of *panC* group VII were not detected in any sample. *panC* group III is considered highly cytotoxic and is representative of *B. cereus* group grown at temperatures of ≥15 °C. The carriage of *nhe* gene (non-hemolytic enterotoxin) characterizes toxin profile F. Strains affiliated to *panC* group VI, which low cytotoxic and grown at ≥5 °C. Toxin profile C is characterized by the presence of *nhe* and *hbl* genes [24] (https://www.tools.symprevius.org/bcereus/english.php; accessed on 17 December 2021).

Naturally contaminated samples were initially pre-screened for the presence of presumptive *B. cereus s.l.* on MYP agar prior to freezing (67.2%, *n* = 43 positive). Recovery after freezing was tested using the selective media test panel, and it resulted in the highest recovery on BA (57.8%, *n* = 37), CH (56.3%, *n* = 36) and MYP (54.7%, *n* = 35) (Figure 6). Appendix A indicates that samples tested negative on MYP before freezing were positive for some of the selective media after freezing. The highest accordance (*n* = 6) for presumptive *B. cereus s.l.* recovery before and after freezing was observed for milk samples of different origin. *panC* group and toxin gene profile combinations of *B. cereus s.l.* detected before and after freezing are provided in Appendix A. In 13 of 64 samples (20.3%), *panC* group and toxin profile combinations were identical before and after freezing. In 12 (18.8%) and 16 samples (25.0%), respectively, *B. cereus s.l.* was detectable either only before or after freezing. In 13 samples (20.3%), different *panC* and toxin combinations were detectable after freezing in comparison to analysis before freezing.

## 4. Discussion

*B. cereus s.l.* is documented among the most prevalent foodborne pathogens, causing one third of food poisoning events in Europe [27].

The presence of *B. cereus s.l*. in food depends mainly on the contamination of the raw material, as well as on recontamination during processing and extrinsic and intrinsic growth conditions during storage. This results in an increased likelihood of disease-relevant concentrations in minimally processed foods consumed either raw or unheated or in inadequately stored extended shelf-life (ESL) products (e.g., in case of cold storage interruption or accidental household refrigerator temperature abuse) [28,29,30,31]. In addition, the availability of nutrients and other extrinsic factors can influence toxin levels formed in the food matrix. In particular, high starch, carbohydrate, vitamin, trace element content, neutral pH and moderate to high water activity have been shown to be associated with increased risk of cereulide formation [32].

The detection of presumptive *B. cereus* requires microbiology-trained personnel and is labor-intensive if samples are comprehensively assessed. Most commonly, detection and confirmation are performed using selective culture media such as MYP agar according to ISO 7932 [21]. In industry, samples are often plated on MYP or PEMBA agar and a further discrimination is pursued. Sample analysis is challenged if a high level of accompanying flora jeopardizes outreads since other microbes will stain the agar yellow due to mannitol consumption. As a result, individual colonies of presumptive *B. cereus s.l.* are missed in the yellow-stained agar, and the sample is often considered false negative by the investigator.

This study focused on the comparison of alternative chromogenic selective nutrient media to identify the best performer for *B. cereus s.l.* detection and enumeration. For this purpose, inclusivity and exclusivity were elicited, and group diversity was determined by using naturally contaminated samples before and after freezing.

The study of *B. cereus s.l.* strains showed an inclusivity of >99% for all media, which is in general very promising. Nevertheless, atypical colony morphologies may occur. The highest rates of atypical β-D-glucosidase negative colonies were observed on BRI (12.7%, *n* = 14)*,* HI (6.4%, *n* = 7) and CH (5.5%, *n* = 6) agar, resulting in a white phenotype. Atypical morphologies were largely related to the milk-derived or soil-derived *panC*-type/toxin profile combination VI/C. These atypical *B. cereus s.l.* phenotypes appear to be niche-specific and may possibly be associated with specific *panC* types with variable exploitability of starch and various carbohydrates in the genetic clade. For instance, *panC* group IV comprises strains isolated from vegetables indicated limited substrate utilization pathways. Furthermore, a sub-branch within *panC* group III showed the least carbohydrate fermentation capacity due to a lack of aryl-6-phospho-β-glucosidase-encoding genes in the genome [33].

Previous studies focusing on agar evaluations also reported ß-D-glucosidase-negative *B. cereus s.l.* colonies on chromogenic *B. cereus* media manufactured by Oxoid or BMC-Biosynth, which is a concern for a proper evaluation [11,12,34]. In contrast, Chon et al. [35] showed increased specificity and selectivity of BRI agar in foods with high background microbial load and particularly recommended this agar for quantitative analysis. In a comparative analysis of BA and BRI agar, these two culture media were clearly superior to conventional culture media, with BRI agar being more efficient and selective for *B. cereus s.l.* isolation in this setting [36]. In a more recent comparison of the standard media MYP, PEMBA, BRI and a novel—yet not commercially listed—chromogenic agar medium (BCCA), atypical colony morphologies were also described on BRI agar (dark blue color) [37]. BCCA was based again on the detection of ß-D-glucosidase comparable to the BRILLIANCE agar and was fortified by polymyxin B (100,000 IU), trimethoprim (10 mg), ceftazidime (16 mg) and egg yolk emulsion (50 mL). This alternative medium seemed to be more selective in comparison to MYP and PEMBA and circumvented the false negative diagnosis of atypically grown presumptive *B. cereus* colonies by additional lecithinase reaction. All this research shows that *B. cereus s.l.* analysis is demanding and that current media are not sufficiently selective to analyze the diversity of the group.

According to literature, the presence of PC-specific or PI-specific PLC is widespread among *B. cereus s.l.* Almost all group isolates were PLC-positive in the literature: 96% [38] and 93% [39] or 100% PLC and 83% PI-PLC positive isolates [40]. The best performer in the detection of PLC reaction mediated by phosphatidyl-inositol (PI) or phosphatidyl-cholin (PC) was BA (98.2%), followed by MYP (97.3%) and CH (95.5%) (Figure 1). Interestingly, four of five inclusivity test strains lacking PLC reaction also showed atypical colony color due to a lack in ß-D-glucosidase on BRI, HI, or CH agar. *Bacillus pseudomycoides* (*panC* group I/toxin profile C) growth was inhibited on MYP, HI and CH (Figure 1). An explanation for this rare atypical observation was provided by Slamti et al. [41], who observed 2% PC-PLC-negative and non-hemolytic test strains due to the absence of PlcR-regulated proteins.

Cross-reactivity for PI-PLC, PC-PLC and β-D-glucosidase was previously observed for *Staphylococcus aureus* and pathogenic *Listeria* [37]. In our study, *Paenibacillus polymyxa* caused PLC cross-reactivity on MYP and BA agar and *L. monocytogenes* grew on BA. ß-D-glucosidase-positive reaction was observed for a broader range of exclusivity test strains (e.g., enterococci, *Listeria*, staphylococci, bacilli, *Microbacterium* and other Gram-negative bacteria) on the tested media (Figure 1 and Figure 3). On BRI, the only agar investigated based on solely one differentiation system (ß-D-glucosidase), several Gram-positive (e.g., *Bacillus*, staphylococci and enterococci) and Gram-negative (e.g., *Enterobacter cloacae, Aeromonas hydrophila* and *Brevundimonas dimenuta*) non-target strains grew despite the addition of polymyxin B in combination with trimethoprim. Lower growth of cocci was observed on BRI in contrast to MYP and HI. However, the proprietary antibiotic mixtures of CH and BA were even superior in selectivity compared to other media.

The detection and differentiation of presumptive *B. cereus s.l.* can be improved by the parallel use of two complementary selective agars, as it is already standard practice in the detection of *L. monocytogenes* [42] and *Salmonella* spp. [43]. The combination of agar media operating on different biochemical principles and characterized by different sensitivity and selectivity (e.g., the highly selective BA or CH with the less selective MYP, BRI, or HI) could allow for a more accurate detection of a broad spectrum of group members in food samples. Since other aerobic spore-formers are also relevant as hygiene indicators in food industry, BRI or HI could be supplemented with egg yolk to detect a broader spectrum of bacilli and improve initial differentiation. Parallel incubation of selective agar plates under mesophilic, psychrophilic, or thermophilic conditions would be recommendable depending on the food type (Figure 7). Incubation at 5–7 °C for the investigation of dairy products may support the assessment of a potential proliferation of bacilli even if the cold chain is maintained [44]. Starch-containing foods as well as herbs and spices have been contaminated with the thermotolerant *B. cytotoxicus*, as shown in previous reports [45,46]. Therefore, thermophilic (≥45 °C) and mesophilic (30 °C) incubation should be considered for these food categories.

In our study, all naturally contaminated samples contained levels of presumptive *B. cereus s.l.* at the limit of detection. In principle, this finding is reassuring, but when assessing the safety of a product throughout the food production chain, including storage to the end of shelf-life, particularly nutrient-rich products contaminated with low levels of *B. cereus s.l.* lacking competitive flora cannot be considered completely safe. On the one hand, one can assume low level contaminations in the case of fresh produce, which, however, can result in rapid multiplication and accumulation of emetic and enterotoxins when temperature deviations occur. Moreover, low contamination levels of highly processed foods do not preclude the presence of the heat-stable and acid-stable toxin cereulide at the time of consumption posing a health risk to the consumer [32]. Naturally contaminated food samples from different manufacturers and batches presented very heterogeneous *B. cereus s.l.* populations. In particular, the diversity of milk isolates between manufacturers was distinctive, which could be attributed to processing methods applied (such as microfiltration and high-temperature treatment) and/or the presence of persister cells in the production environment (Figure 5, Appendix A).

Naturally contaminated samples were pre-screened for the presence of presumptive *B. cereus s.l.* prior to freezing. Recovery after freezing was tested using the selective media test panel and resulted in the highest recovery on BA (57.8%), CH (56.3%) and MYP (54.7%) (Figure 6). Identical *panC* group and toxin gene combinations before and after freezing were detected in 20.3% of samples. Sampling before and after freezing revealed shifts in *panC* groups and toxin gene profiles, but within samples from the same producer the distributions were consistent. In 15.6% of samples, divergent *panC* and toxin combinations were detected after freezing. This phenomenon can be explained by the non-uniform distribution of *B. cereus s.l.* contamination at the detection limit (Poisson distribution) and by the influence of matrix components during initial testing [47]. Group species and their toxins may be bound to lipid globules (e.g., in the case of dairy products) and only become detectable following rougher digestion after more stringent sample treatment process procedures, e.g., using such as beads beating [48,49]. In our investigation, freezing samples resulted in the detection of an extended spectrum of *panC* and toxin profile combinations.

The predominant *panC*/toxin profile combination among target strains and naturally contaminated sample isolates was IV/A (21.8% and 33.0%, respectively), followed by VI/C (17.3% and 10.4%), III/F and II/F (21.7% and 10.4% in naturally contaminated samples) and III/D (10.9% target strains) (Figure 5 and Appendix A).

Phylogenetic groups II, III and IV comprise moderately to highly cytotoxic strains, most likely posing a potential health risk. In addition, *panC* group III strains may carry the *ces* gene encoding for emetic toxin cereulide [25]. *B. cereus s.l.* strains assigned to *panC* group VI were often isolated from raw milk (target strain set) and were highly abundant among heat-treated milk samples (Appendix A) [50].

Recently, the connection of biopesticidal *B. thuringiensis* strains to foodborne outbreaks in France was investigated. In 39% of outbreaks, *B. thuringiensis panC* group IV was suspected to be the causative organism [51]. In our study, *panC* group IV was highly abundant among isolates from salads, vegetables, herbs and spices that may also include biopesticidal *B. thuringiensis* strains. Furthermore, *cytK*-2 was highly abundant among *panC* group IV strains [18,52]. This is concordant with our results as we identified *cytK*-*2* highly abundant in *panC*/toxin gene profile combination IV/A (21.8% and 33.0% among target strains and sample isolates). Since other studies have found the use of *B. thuringiensis* biopesticides to be safe or of low risk to public health [53,54], future research should address the contribution of extensively used biopesticide strains to the contamination of raw materials, such as vegetables and fresh produce processed into ready-to-eat foods.

## 5. Conclusions

This study dealt with culture-based *B. cereus s.l.* diagnosis, which is especially practiced in routine analysis. We tested a selective media panel using test strains and naturally contaminated samples at the detection limit, which is relevant for practice. The results show that it is necessary to include more than one selective medium in the analysis, comparable to *Listeria monocytogens* and *Salmonella* diagnostics in food and animal feed. In order to be able to make a statement about contamination with presumptive *B. cereus s.l.* at all, it is recommended to perform, e.g., PCR, FTIR or MALDI-based confirmation and subtyping (e.g., *panC* and toxin gene profiling) and to assess growth behavior (e.g., psychrotolerance) (Figure 7).

## Figures and Tables

**Figure 1 foods-11-00288-f001:**
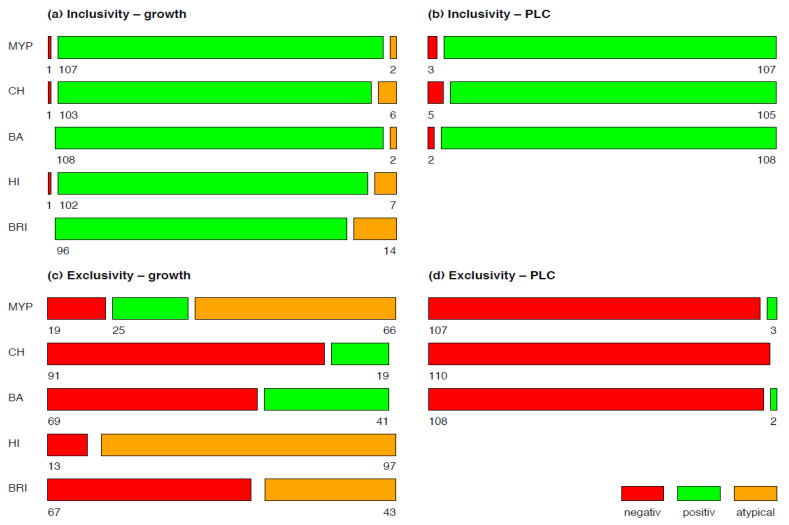
Growth (**a**,**c**) and phospholipase C reaction (**b**,**d**) of inclusivity (*n* = 110) and exclusivity (*n* = 110) test strains. Negative is no-growth, positive is typical growth and atypical is not presumptive *Bacillus cereus sensu lato* morphology on selective agar media. Abbreviations: PLC—phospholipase C; MYP—mannitol egg yolk polymyxin agar; CH—CHROMagar™ *B. cereus*; BA—BACARA^®^ agar; HI—ChromoSelect *Bacillus* agar; BRI—Brilliance™ *B. cereus* agar.

**Figure 2 foods-11-00288-f002:**
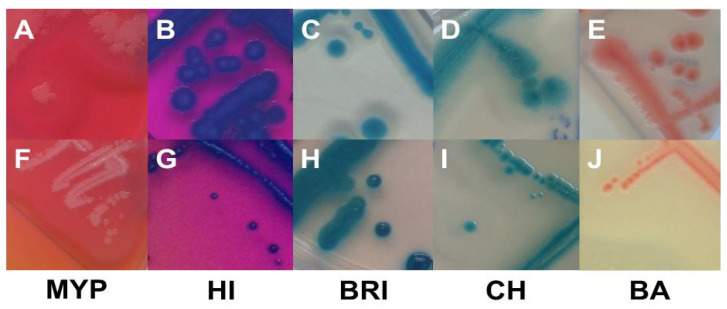
Typical (**A**–**E**) and atypical colonies (**F**–**J**) of *Bacillus cereus sensu lato* on MYP agar (A (BCG 6), F (BC 66)), ChromoSelect *Bacillus* agar (B (BC 30), G (BC 20)), Brilliance™ *B. cereus* agar (C (BC 2), H (BC 19)), CHROMagar™ *B. cereus* (D (BC 63), I (BC 2)) and BACARA^®^ agar (E (BC 50), J (BC 34)). Bluish-green colonies are the result of β-D-glucosidase reaction. Precipitation zones surrounding typical colonies are caused by phospholipase C reaction, while lack of mannitol fermentation results in pink background. Abbreviations: MYP—mannitol egg yolk polymyxin agar; HI—ChromoSelect *Bacillus* agar; BRI—Brilliance™ *B. cereus* agar; CH—CHROMagar™ *Bacillus cereus*; BA—BACARA^®^ agar.

**Figure 3 foods-11-00288-f003:**
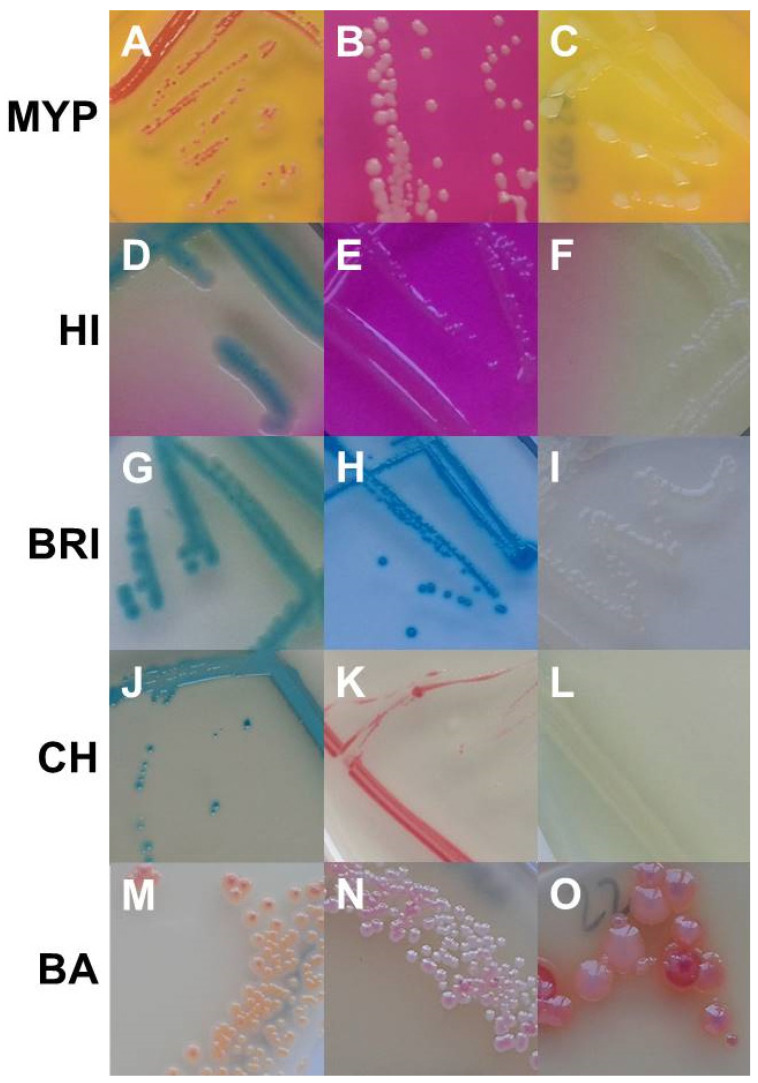
Growth of non-target organisms on *Bacillus cereus* selective media (from upper left to lower right). MYP agar: (**A**)—*Serratia marcescens* (EGN 54); (**B**)—*Brochothrix thermospacta* (EGP 2); (**C**)—*Bacillus stratosphericus* (BG 28). ChromoSelect *Bacillus* agar: (**D**)—*Aeromonas hydrophila* (EGN 5); (**E**)—*Acinetobacter baumannii* (EGN 2); (**F**)—*Citrobacter freundii* (EGN 9). Brilliance™ *B. cereus* agar: (**G**)—*Staphylococcus sciuri* (EGP 13); (**H**)—*Serratia marcescens* (EGN 54); (**I**)—*Pseudomonas fluorescens* (EGN 45). CHROMagar™ *B. cereus*: (**J**)—*Morganella morganii* (EGN 39); (**K**)—*Enterococcus faecalis* (EGP 5); (**L**)—*Providencia rettgeri* (EGN 42). BACARA^®^ agar: (**M**)—*Staphylococcus haemolyticus* (EGP 11); (**N**)—*Staphylococcus chromogenes* (EGP 9); (**O**)–*Listeria monocytogenes* (EGP 22). Abbreviations: MYP—mannitol egg yolk polymyxin agar; HI—ChromoSelect *Bacillus* agar; BRI—Brilliance™ *B. cereus* agar; CH—CHROMagar™ *Bacillus cereus*; BA—BACARA^®^ agar.

**Figure 4 foods-11-00288-f004:**
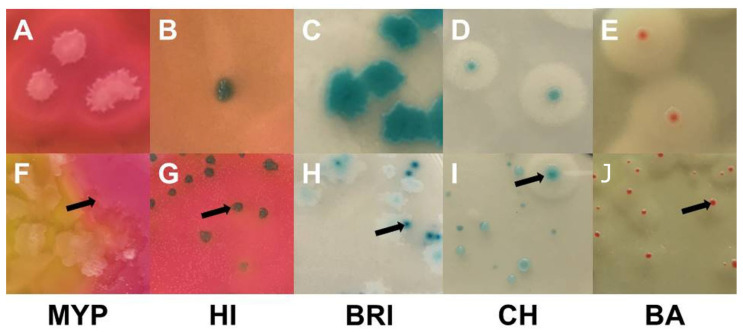
Examples of typical *Bacillus cereus sensu lato* colonies obtained by sampling of naturally contaminated food ans milk samples. Demarcated colonies with typical morphology (top row): (**A**)—MYP agar (ESL-milk); (**B**)—HI, ChromoSelect *Bacillus* agar (ESL-goat milk); (**C**)—Brilliance™ *B. cereus* agar (ESL-milk); (**D**)—CHROMagar™ *B. cereus* (ESL-milk); (**E**)—BACARA^®^ agar (ESL-milk). Colonies masked by high growth of background flora and atypical morphologies (bottom row; arrows point on typical *B. cereus* colonies): (**F**)—coalescing *B. cereus* colonies surrounded by mannitol-positive background-flora (*B. licheniformis*) on MYP agar (dried fish snack); (**G**)—mixed culture on ChromoSelect *Bacillus* agar, growth of mannitol-positive background-flora (*Staphylococcus* spp.) intersparsed with typical colonies (raw milk); (**H**)—atypical light colonies with weak β-D-glucosidase activity and typical colonies on Brilliance™ *B. cereus* agar (ESL-milk); (**I**)—atypical PLC-negative and weakly β-D-glucosidase positive colonies lacking the distinctive halo together with typical colony on CHROMagar™ *B. cereus* (Chinese water spinach); (**J**)—atypical small colonies with weak PLC acitivity on BACARA^®^ agar (dried fish snack). Abbreviations: MYP—mannitol egg yolk polymyxin agar; HI—ChromoSelect *Bacillus* agar; BRI—Brilliance™ *B. cereus* agar; CH—CHROMagar™ *B. cereus*; BA—BACARA^®^ agar.

**Figure 5 foods-11-00288-f005:**
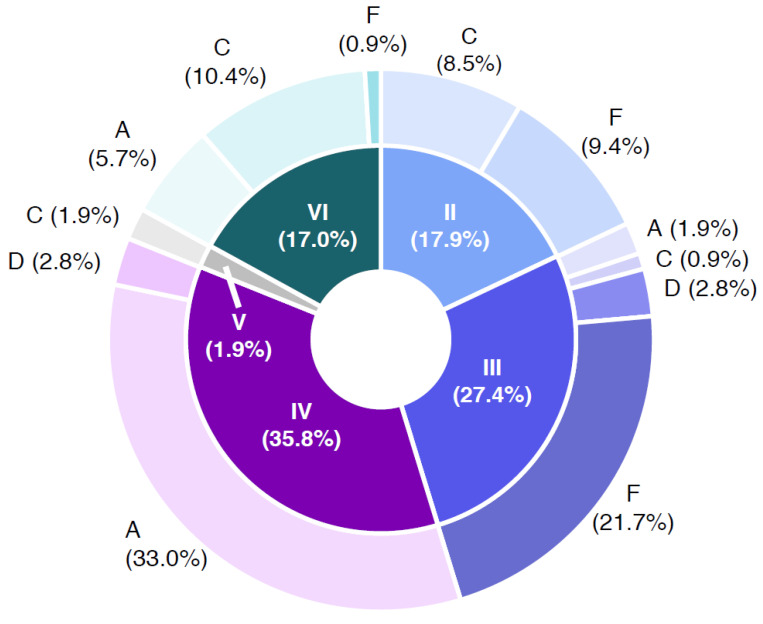
Combinations of *panC* groups (II–VI) and toxin gene profiles (A–F) of *Bacillus cereus sensu lato* isolates obtained from 64 naturally contaminated samples. Abbreviations: A—toxin profile A (*nhe*+, *hbl*+ and *cytK+*); C—toxin profile C (*nhe+* and *hbl+*), D—toxin profile D (*nhe+* and *cytK+*); F—toxin profile F (*nhe*+); II—*panC* group II (cytotoxic, growth ≥7 °C); III—*panC* group III (cytotoxic-highly cytotoxic, growth ≥15 °C); IV—*panC* group IV (highly cytotoxic, growth ≥10 °C); V—*panC* group V (low cytotoxic, growth ≥8 °C); VI—*panC* group VI (non or low cytotoxic; growth ≥5 °C).

**Figure 6 foods-11-00288-f006:**
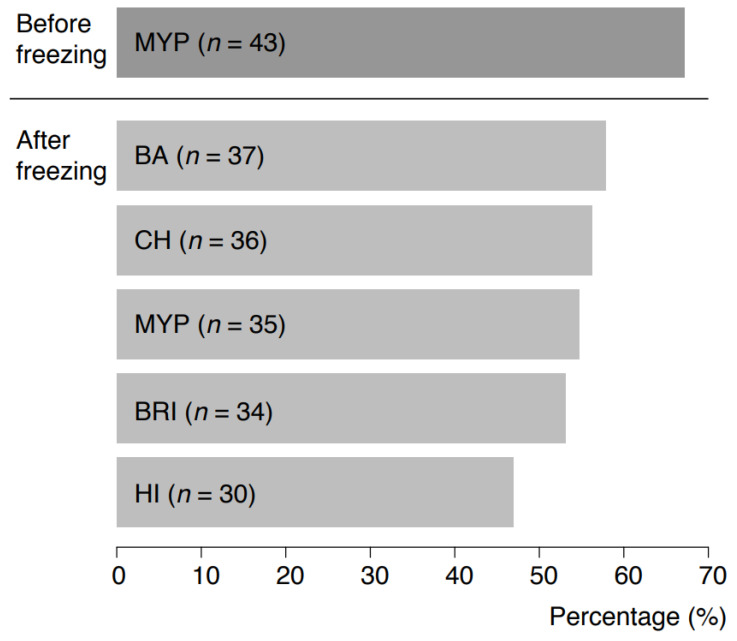
Detectability of *Bacillus cereus* sensu lato in 64 naturally contaminated food samples before freezing on MYP agar and after freezing on MYP agar and chromogenic media. Abbreviations: MYP—mannitol egg yolk polymyxin agar; BA—BACARA^®^ agar; CH—CHROMagar™ *Bacillus cereus*; BRI—Brilliance™ *B. cereus* agar; HI—ChromoSelect *Bacillus* agar.

**Figure 7 foods-11-00288-f007:**
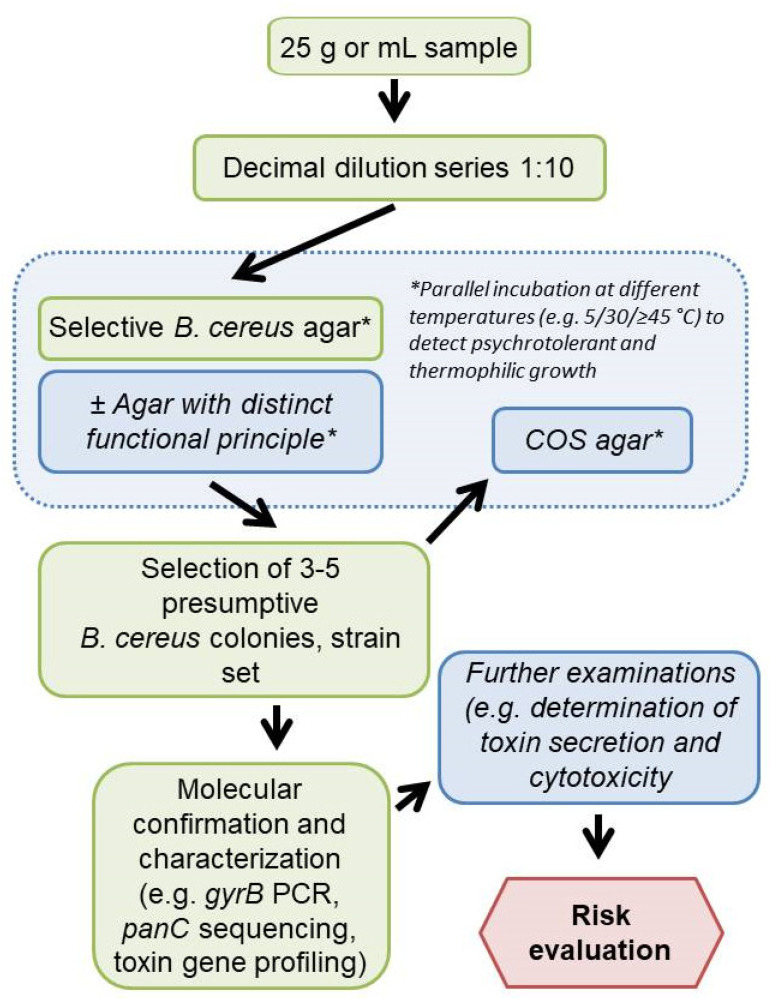
Proposed workflow for *Bacillus cereus sensu lato* analysis performable in routine food analysis (Italic—optional steps). COS, Columbia agar plus 5% sheep blood. *, parallel incubation at different temperatures to detect psychrotolerant and thermophilic growth.

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
