# Peer review of "Performance Testing of Bacillus cereus Chromogenic Agar Media for Improved Detection in Milk and Other Food Samples"

_foods, 2022, doi:10.3390/foods11030288_

Round 1

Reviewer 1 Report

Overall, a very interesting and correctly written manuscript. My only remarks result from the fact that the article is not adapted to the requirements of the authors, among other things, the incorrect writing of the cited literature, or the lack of a separate "conclusions" section at its end. After the manuscript has been corrected and adapted to the requirements of the editorial office. Minor comments are included in the attached pdf file. 

Author Response

AUTHORS (AU): Thank you very much for your thorough review of the manuscript. We have now revised the manuscript, according to your kind suggestions. The changed sections have been highlighted in yellow.

The formatting of the references has been adjusted according to your remark.

Reviewer 2 Report

The article by Fuchs et al. on Performance testing of selective agar media and molecular characterization of Bacillus cereus group members for improved detection in milk and other food samples" is well written and contains interesting information. The pictures are fine to read, clear and easy to understand. The results presented are based on experiments that are well done. On the other hand, the formatting of the article does not comply with the requirements of the journal. 
Overall, I consider the article to be thoroughly described. In particular I recommend the author to check the text formatting (e.g. ß-D-glucosidase - D should be in capital letters; "ml" in Figure 7).

Author Response

AUTHORS (AU): Thank you very much for your thorough review of the manuscript. We have now revised the manuscript, according to your kind suggestions. The changed sections have been highlighted in yellow.

The formatting of references was adjusted to the journals requirements. 

The uniform spelling of ß-D-glucosidase was checked according to your suggesion.

Figure 7: mL was corrected

Reviewer 3 Report

General comments:

the article “Performance testing of selective agar media and molecular characterization of Bacillus cereus group members for improved detection in milk and other food samples” (by Fuchs et al., Foods) reports on a new method development to specifically detect species members of B. cereus groups in foods.

  • The wok is of interest for the “microbiological methods of bacteria investigation”, but as the title is, it is not clear whether the authors focused on a new selective agar medium and they used the molecular characterization to confirm the goodness of the agar medium, or if the strategy combines together agar media and molecular characterization. In the latter case, the method is not time-saving, thus not particularly attractive to be used as alternative to the classical cultivation approach based on colony isolation and molecular processing. The title should be rewritten for clarity.
  • The abstract is extremely long. The initial part, L18-21, is useless and can be deleted.
  • L41-45. A reader would expect a clear indication on how samples have to be treated before species detection. Please rewrite this part providing a clear methodology, avoiding to be vague on the “Depending on sample properties and expected contaminants, incubation of agar plates at different temperatures and/or analysis before and after freezing may be employed”, be specific and synthetic.
  • I don’t feel qualified to judge the English style, the work seems to be well written and comprehensive for the language, the problem is the unclear content.
  • L50-59. The two paragraph could be merged into a single one.
  • In this context, “infant formulas” seems more appropriated.
  • In general, it is extremely long and unfocused in several general parts. Please reduce the length of this paragraph focussing on the methodologies for detection. Reduce the negative effects of B. cereus groups, they are well known. I suggest to reduce the length of 30-40%. As is, Introduction, tires the readers.
  • I found several problems following the choice of the strains. In general, when developing new microbiological methods, firstly it is necessary to evaluate specificity with all target strains of the identification and, secondly, validate the system with non target strains, closely related to the target ones. In this work, the authors used Gram-negative bacteria, which are unrelated to the target organisms. To this purpose, figures do not provide any indication on the strains used and, in my opinion, this represents a high limitation to the work. Please consider this as a major point of the work that needs particular attention.
  • It is also difficult to follow the sampling at paragraph 2.1.3.

Author Response

Reviewer 3 (RE): The article “Performance testing of selective agar media and molecular characterization of Bacillus cereus group members for improved detection in milk and other food samples” (by Fuchs et al., Foods) reports on a new method development to specifically detect species members of B. cereus groups in foods.

AUTHORS (AU): Thank you very much for your thorough review of the manuscript. We have now revised the manuscript, according to your kind suggestions. The changed sections have been highlighted in yellow.

RE 3: The wok is of interest for the “microbiological methods of bacteria investigation”, but as the title is, it is not clear whether the authors focused on a new selective agar medium and they used the molecular characterization to confirm the goodness of the agar medium, or if the strategy combines together agar media and molecular characterization. In the latter case, the method is not time-saving, thus not particularly attractive to be used as alternative to the classical cultivation approach based on colony isolation and molecular processing. The title should be rewritten for clarity.

AU: thank you for your suggestion, we changed the title to “Performance testing of Bacillus cereus chromogenic agar media for improved detection in milk and other food samples.”

The focus of the workflow we followed was to be more precise in the differentiation of B. cereus s.l.. Therefore, we compared commercially available selective media and tested the performance. Chromagar and Bacara agar were time-saving in the first step of evaluation of presumptive B. cereus colony morphologies as atypical colonies and background flora were limited. The idea of combining molecular subtyping by panC grouping and toxin gene profiling was to be more precise in the assessment of the isolate properties of B. cereus s.l. with regards to phylpgenetic groups, predicted growth preferences and pathogenic pontential.

RE 3: The abstract is extremely long. The initial part, L18-21, is useless and can be deleted.

AU: thank you for your suggestion. We reduced the length of the abstract according to your suggestion.

RE 3: L41-45. A reader would expect a clear indication on how samples have to be treated before species detection. Please rewrite this part providing a clear methodology, avoiding to be vague on the “Depending on sample properties and expected contaminants, incubation of agar plates at different temperatures and/or analysis before and after freezing may be employed”, be specific and synthetic.

AU: the abstract was revised according to your suggestion.

RE3: I don’t feel qualified to judge the English style, the work seems to be well written and comprehensive for the language, the problem is the unclear content.

AU: we revised the manuscript according to your suggestion to be clear and improved the readability.

RE 3: L50-59. The two paragraph could be merged into a single one.

AU: thank you the paragraph was merged.

RE 3: In this context, “infant formulas” seems more appropriated.

AU: thank you was corrected.

RE3: In general, it is extremely long and unfocused in several general parts. Please reduce the length of this paragraph focussing on the methodologies for detection. Reduce the negative effects of B. cereus groups, they are well known. I suggest to reduce the length of 30-40%. As is, Introduction, tires the readers.

AU: according to your suggestion we reduced several passages of the manuscript.

RE3: I found several problems following the choice of the strains. In general, when developing new microbiological methods, firstly it is necessary to evaluate specificity with all target strains of the identification and, secondly, validate the system with non target strains, closely related to the target ones. In this work, the authors used Gram-negative bacteria, which are unrelated to the target organisms. To this purpose, figures do not provide any indication on the strains used and, in my opinion, this represents a high limitation to the work. Please consider this as a major point of the work that needs particular attention.

AU: Thank you with recommendation. We did not develop new media, the chromogenic media are available at the moment and we tried to figure out which chromogenic agar is the best performer with respect to productivity, selectivity and sensitivity. We would suggest to use two complementing agar types to avoid false negative results. Overall, the background flora is not totally inhibited by supplements and if the have an advantage in growth B. cereus could be masked and not detected. Therefore, we also included also Gram-negative strains to get a complete picture of the performance of each medium. We have included in figure 2, 3, 6 the strain identities according to supplement table 2 and 3.

RE3: It is also difficult to follow the sampling at paragraph 2.1.3.

AU: the paragraph was revised.

Round 2

Reviewer 3 Report

The authors modified the paper considering all suggestions. The rebuttals to the most critical points were satisfactory. I do not have other comments. The paper seems to be ready to be published.